# Efficacy and Safety Ablation Index-Guided High-Energy Linear Ablation for Persistent Atrial Fibrillation: PVI Plus Linear Ablation of Mitral Isthmus and Posterior Box Isolation

**DOI:** 10.3390/jcm12020619

**Published:** 2023-01-12

**Authors:** Xi Li, Tao Liu, Bo Cui, Jinlin Zhang, Yanhong Chen, Gang Wu

**Affiliations:** 1Department of Cardiology, Renmin Hospital of Wuhan University, Wuhan 430060, China; 2Cardiovascular Research Institute, Wuhan University, Wuhan 430060, China; 3Hubei Key Laboratory of Cardiology, Wuhan 430060, China; 4Department of Cardiology, Wuhan Asian Heart Hospital, Wuhan 430060, China

**Keywords:** persistent atrial fibrillation, catheter ablation, pulmonary vein isolation, linear ablation, ablation index, high-power, posterior wall isolation

## Abstract

Background: For patients with persistent atrial fibrillation (AF), whether linear ablation should be performed remains controversial, and the efficacy and safety for ablation index (AI)-guided high-energy linear ablation of mitral isthmus (MI) and left atrial (LA) posterior box isolation is still unclear. The aims of this study were to assess the feasibility and clinical success rate of pulmonary veins isolation (PVI) combined with linear ablation of LA roof and posterior inferior (posterior wall isolation) and MI compare with the PVI-alone method in patients of persistent AF. Methods and results: 362 consecutive persistent AF patients were enrolled from two electrophysiology centers. A total of 200 cases were in PVI-plus group and 162 cases were in PVI-alone group. The PVI-alone group received wide circumferential isolation of both ipsilateral pulmonary veins. PVI combined with linear ablation of left atrial posterior wall isolation (LAPWI)and MI were performed in the PVI-plus group. The primary study end point was the first recurrence of an atrial arrhythmia. After 24 months, freedom from the primary endpoint was achieved in 73.5% of the patients in the PVI-plus group and 62.5% in the PVI-alone group (hazard ratio = 0.62, 95% confidence interval: 0.43–0.91, log rank *p* = 0.012). The procedure-related complication rates were 2.5% in PVI-plus group and 1.9% in PVI-alone group (*p* = 0.808). Conclusion: In this study, the ablation strategy of ablation (PVI plus linear ablation of mitral isthmus and posterior box isolation) was feasible and safe for persistent AF patients. Compared with the PVI-alone method, it improved outcomes in patients with persistent AF.

## 1. Introduction

Pulmonary vein isolation (PVI) is the cornerstone of catheter ablation of atrial fibrillation (AF) [1]. For patients with persistent AF, the mechanisms are relatively complex, and additional ablation was recommended [2,3,4]. However many studies have shown that adding ablation did not significantly improve the success rate of AF ablation [5,6,7,8]. Currently, the optimal ablation strategy for persistent AF remains controversial. The ablation strategy for persistent AF adds linear ablation of the mitral isthmus (MI) and electrical isolation of left atrial (LA) posterior wall (PW) using ablation index (AI)-guided high-power ablation (no fragmented potential ablation was performed at the index procedure) was not reported. In this study we report the efficacy and safety of this strategy to treat persistent AF, compared with the PVI-alone method.

## 2. Methods

Trial management and design

This is a retrospective analysis, and patients with persistent AF from May 2019 to December 2019 were included in our study. Local ethics review committees at each center approved the study. All methods were performed in accordance with the relevant guidelines and regulations. All the patients met the diagnostic criteria of AF and provided written informed consent to undergo RFCA. Persistent AF was defined as a sustained episode lasting ≥7 days [9,10]. Exclusion criteria included valvular AF, history of cardiac surgery, dialysis or heart failure (LVEF < 30% and New York Heart Association classification III or IV), left atrial thrombus and age ≥80 years. 

Pre-procedural

Before the ablation procedure, blood tests, transthoracic echocardiogram, transesophageal echocardiography, cardiac contrast-enhanced CT, 12-lead electrocardiogram (ECG) and 24 h electrocardiogram (Holter) were obtained. The absence of any LA thrombi was confirmed by transesophageal echocardiography.

General principles of ablation:

The procedure was performed under general anesthesia using the CARTO-3 mapping system (Biosense Webster, Diamond Bar, CA, USA). A decapolar coronary sinus (CS) catheter was introduced via left subclavian vein, and then double transseptal punctures were performed under fluoroscopic guidance. Intravenous heparinization was performed when two atrial septal punctures were completed. Two catheters were introduced via the femoral vein: (1) an irrigated-tip radiofrequency (RF) ablation catheter (ThermoCool SmartTouch STSF; Biosense Webster); and (2), a multipolar mapping catheter (Pentaray; Biosense Webster). The target activated clotting time was 300 to 350 s. A LA 3D electroanatomical map using the CARTO 3 System was performed using a Pentaray catheter. Point-by-point ablation was performed in power-control mode (temperature 43 °C; saline irrigation 15 mL/min). Ablation with high power and RF current was delivered for 10–30 s (power 45 W). Automated lesion annotation was performed using the VisiTag module (catheter stability range of motion 3 mm for 5 s; force ranges 10–15 g for 70% of time; Biosense Webster). The ablation index (AI) targets were 400 for the LA roof and posterior inferior line, PVI (ridge 500; inferior 450; superior 450; posterior 400), and 450 for the MI ablation line. Patients with adverse events were defined as death, symptomatic stroke, cardiac tamponade, phrenic nerve paralysis, femoral arteriovenous fistula, femoral pseudoaneurysm, atrioesophageal fistula, infection, heart failure, and pulmonary vein stenosis.

The strategy of ablation:

PVI-alone group ablation

The protocol procedure included wide circumferential isolation of both ipsilateral pulmonary veins (Figure 1A) with verification entrance and exit block using point-by-point irrigated-tip radiofrequency catheter ablation according to clinical standards [9].Electrical cardioversion: If AF has not been terminated after the completion of PVI, the electrical cardioversion (bi-phase 150–200 J) would be performed.

PVI-plus group ablation

3.PVI: PVI was performed and verified as previously described.4.MI linear ablation: linear ablation was performed across the MI between the left inferior PV and mitral annulus. AI-guided high-power ablation was used with the ablation energy of 45 W and the target AI of 450 (Figure 1B, Appendix A).5.PWI: LA roof linear ablation was performed across the LA roof joining the two superior isolated PV. LA posterior inferior linear joined the two inferior isolated PV. AI-guided high-power ablation was used with the ablation energy of 45 W and the target AI of 400.6.Electrical cardioversion: If AF has not been terminated after the completion of PVI and linear ablation, the electrical cardioversion (bi-phase 150–200 J) would be performed to stop AF.7.If AF was converted to AFL (atrial flutter) during ablation, reentrant circus was determined by high-density activation mapping, then linear ablation of the key isthmus was performed, including the tricuspid isthmus. Whenever AF terminated to one or more atrial tachycardias (ATs), these were targeted for ablation until sinus rhythm (SR)was achieved.

Ablation line block examination:8.MI ablation line conduction block: Bidirectional block established by pacing in the LAA and high-density activation mapping with the Pentaray catheter (Figure 1C). Ablation was performed within the CS vein if the MI ablation line was not blocked [11,12] (Figure 2).9.PW conduction block: After the roof and posterior inferior ablation lines had been completed, the PWI was confirmed with either the electrical silence of the LA posterior wall (Figure 3B), or a conduction block when pacing the LA post wall. (Figure 3C). If the LA posterior wall was not isolated, additional ablations were performed until PWI was achieved.

Inducing the tachycardia: 

After the lesion set was completed in PVI-plus group, burst pacing was delivered from proximal CS at decreasing pacing cycle lengths down to 200 ms (or to the local refractory period) to induce atrial tachycardia (AT). If atrial flutter was induced, we used high-density activation mapping with the Pentaray catheter, and then ablation was performed until SR was achieved. If AF was induced, the electric cardioversion or intravenous ibutilide cardioversion was applied to terminate it.

Postprocedural management

After ablation, all patients were treated with proton pump inhibitors for 4 weeks to prevent esophageal injury. During the blanking period, the use of anti-arrhythmic drugs (AADs) was allowed. However, discontinuation of AADs was strongly recommended after the blanking period [13].

Follow-up

Follow-up at 1, 3, 6, 12, 24 months and beyond was performed. A 12-lead electrocardio gram (ECG) was routinely reviewed 1 month after the operation. A 7-days Holter ECG and transthoracic echocardiogram was acquired at 3, 6, 12, 24 months. In addition to the scheduled follow-up, the telephone follow up has been carried out by dedicated follow-up commissioners. Patients were strongly recommended to visit a healthcare provider if they felt symptoms possibly due to an arrhythmia or noticed any irregularity of their peripheral pulse by self-measurement. An ECG was performed at every additional visit, and cases with symptoms or findings suggestive of recurrence underwent Holter ECG monitoring. Arrhythmia recurrence was defined as any episode of AF/AT lasting > 30 s after a 3-month blanking period post-ablation [9]. 

Statistical analysis 

The continuous data were tested for normal distribution. If normal distribution was found, data were expressed as the mean ± SD and analyzed by Student’s t test. If abnormal distribution was found, data were presented as the median with interquartile range and compared by the Welch’s t-test. Categorical variables are given as absolute number (percentage). Freedom from atrial arrhythmia recurrence during follow-up of up to 24 months was analyzed by using survival analysis for cumulative event rates including Kaplan–Meier estimates and Cox regression for calculation of odds ratios. Odds ratios are presented with 95% Cl. To obtain the optimal cut-off values for predictors of AF recurrence, Youden index was calculated as sensitivity + specificity − 1. The value for the maximal Youden index was considered as the optimal cut-off point. A *p*-value of *p* < 0.05 was deemed statistically significant. All statistical analysis was done using SPSS (Version 26, IBM Corp., Armonk, NY, USA).

## 3. Results

Patient characteristics

A total of 362 consecutive patients were enrolled between March 2019 to December 2019. A total of 200 cases were in the PVI-plus group, and 162 cases were in PVI-alone group. Baseline characteristics were summarized in Table 1; demographic characteristics were similar between two groups. 

Procedure Outcome

The procedural outcome was shown in Table 2. Successful PVI was achieved 100% in two groups. In the PVI-plus group, the total procedure time and fluoroscopy time were longer than the PVI-alone group (*p* < 0.001).

In the PVI-plus group, the immediate success rate of MI line conduction block during the procedure was 89.5% (179/200). A total of 76% (152 of 200) patients received ablation in the CS vein to achieve MI block. In 21 patients, MI conduction block was not achieved despite of repeated ablations at the MI and CS vein during the procedure. The success rate of PW conduction block during the procedure was 96% (192/200). A total of 28% (56/200) of patients conversed to SR. A total of 26 of 56 patients were directly converted to SR during ablation. Additionally, 30 of 56 patients converted to AFL during ablation, and finally converted to SR by further ablation of AFL using activation mapping. 

In the PVI-alone group, 8% (13/162) patients were directly converted to SR during ablation.

Complications 

In the PVI-plus group, one patient had ischemic stroke post operation, who recovered without severe morbidity after treatment. Four patients experienced a femoral pseudoaneurysm that did not require intervention. 

In the PVI-alone group, 3 patients experienced femoral pseudoaneurysm that did not require intervention. 

No death, cardiac tamponade of clinical significance, PV stenosis, phrenic nerve palsy, and atrial-esophageal fistula was presented peri-operation (Table 3).

Clinical Outcome

At the end of follow-up, AF/AT events occurred in 53 (26.5%) and 61 (37.7%) patients from the PVI-plus and PVI-alone groups, respectively (Figure 4). The AF/AT recurrence rate was significantly lower in the PVI-plus group than that in the PVI-alone group (hazard ratio = 0.62, 95% confidence interval: 0.43–0.91, log rank *p* = 0.012) after the 24-month follow-up period.

In the PVI-plus group, AF recurred in 48 patients and AT recurred in 5 patients. A total of 60.4% (32/53) of patients with AF/AT recurrence had a second procedure. During the re-ablation, recovered pulmonary veins potentials were observed in 9 patients, recovered MI conduction were found in 20 patients, and failed PWI were confirmed in 12 patients.

In the PVI-alone group, AF recurred in 59 patients and AT recurred in 2 patients. A total of 45.9% (28/61) of patients with AF/AT recurrence underwent re-ablation. During the second ablation, recovered pulmonary veins potentials were observed in 19 patients.

Risk factors predicting AF recurrence after index ablation

In univariate cox regression analysis, the risk factors predicting AF recurrence in PVI-plus group included AF duration (HR: 1.02, 95%CI: 1.01–1.02, *p* = 0.001) and LA anteroposterior diameter (LAd) before catheter ablation (HR: 3.19, 95%CI: 1.53–6.67, *p* = 0.002). There was no conversion to SR during ablation (HR: 2.61, 95%CI: 1.17–5.79, *p* = 0.019). In the multivariate cox regression analysis, AF duration (HR: 1.01, 95%CI: 1.00–1.02, *p* = 0.012), LAd (HR: 2.52, 95%CI: 1.22–5.20 *p* = 0.012) and no conversion to SR during ablation (HR: 2.45 95%CI: 1.10–5.44, *p* = 0.028) remained the independent predictors of AF recurrence. (Table 4). The cutoff value of LAd before catheter ablation predicting AF recurrence was 4.5 cm (AUC = 0.61, *p* = 0.023) (Appendix A), and the cutoff value of AF duration was 24.5 months (AUC = 0.71, *p* < 0.01) (Appendix A).

## 4. Discussion

Main findings

In our study, persistent AF patients were treated with an AI-guided high-power ablation strategy (PVI combined with linear ablations of MI, PWI), which had a relatively high success rate versus the PVI-alone group. There was no difference in adverse events; there was an approximate 10% increase in procedure duration and ablation time for PVI-plus versus PVI-alone. The LAd ≥ 4.5 cm, AF duration ≥ 24.5 months and no conversion to SR were independent factors predicting AF/AT recurrence after index ablation. 

Strategy of ablation

In persistent AF, non-pulmonary vein-originated triggers play important roles in the initiation and maintenance of AF, and PVI alone failed to achieve satisfactory clinical outcomes [14]. It is reported that additional LA linear ablation is helpful to improve the success rate [15]; the key of LA linear ablation is to ensure the bidirectional block (BDB) of ablation lines. The failure in linear block and the high linear conduction recovery rate are the most important limitations of this approach that attribute to the recurrence of ATs e [16]. AI, also known as VISITAG SURPOINT, is a novel lesion-quality marker that improves outcomes in radiofrequency (RF) catheter ablation of AF [17,18]. At the same time, high power ablation has also been proved to be effective in the AF ablation [19,20]. We postulated that high power could help in achieving the transmural tissue injury, and that AI-guided ablation could monitor the stability of the catheters, both of which could improve the success rate of complete line block. We have found that the first pass of PVI and linear ablation is relatively high, compared to conventional ablation settings (Table 2). Further, we tend not to do complex fractionated atrial electrograms (CFAEs) ablation in the index procedure, because too much CFAEs ablation might give rise to iatrogenic AFL [21].

Linear ablation of MI

Although linear ablation of MI presents certain challenging [22], contact pressure and catheter tip orientation are essential in the depth and size of the lesion. In our study, the rate of MI block was 89.5%, which is satisfactory. It may be because that we use high power and the AI-guided module to ensure lesion size.

To achieve bidirectional block of the ablation line and reduce ablation complications, it is also important to understand the anatomical structure before ablation [23]. We thus performed routine cardiac CTA to identify pouch on the planned ablation area (from mitral valve to Left inferior pulmonary vein) (Figure 1D), which helped a lot to avoid possible pop or unnecessary LA injury. If there is an obvious pouch between the mitral valve and the left lower pulmonary vein, we tried to bypass it during the operation, reducing ablation power to 35 W to reduce the risk of pericardial tamponade. CS ablation is also important to achieve the MI block [24]. In our study, 76% of patients needed additional ablation in the CS, where the power was set to 25 W, using vector monitoring to determine the direction of the catheter tip orientation at all times. Ablation in the CS was performed until the activation sequence of CS changes (Figure 2C). However, there were still a few patients for who block MI could not be achieved during operation. If MI and CS ablation cannot block the ablation line of MI, the Marshall ethanol infusion could be an additional ablation method to improve the block rate [25].

The PWI

The posterior wall of the LA has also been shown to serve as one of the non-PV foci in the initiation of AF [26]; electrical isolation of LA posterior wall can improve the success rate of ablation of PAF [27]. We did the LA roof and posterior inferior linear ablation to complete the isolation of the LA posterior wall. The linear block at the LA roof and posterior inferior can be achieved in most patients, but may be challenging in some patients. The LA roof is the most common site of posterior wall reconnection [28], as we found in the secondary ablation of the recurrent patients. When the LA roof line cannot be blocked by ablation during operation, the epicardial conduction via the septopulmonary bundle may have to be taken into account [29].

AF termination 

It is demonstrated that AF termination during catheter ablation is a strongest predictor of the clinical outcomes in patients with longstanding persistent AF [30]. In our study, 56 of 200 patients were converted to SR by catheter ablation, who had a higher success rate of SR restoration compared to those recover to SR by electrical cardioversion during the 24-month follow-up period (Figure 5). Furthermore, in the index procedural, 23 patients were switched from AF into AFL during ablation and finally recovered to SR by further high-density mapping and ablation. Thus, it is reasonable to consider that the conversion from AF to AFL in-procedural is a favorable sign for predicting the AF termination by catheter ablation.

Recurrence predictors before ablation

Predictors of recurrence before ablation are important in patients with atrial fibrillation, and we can more accurately screen patients, set realistic patient expectations, and predict the success of atrial fibrillation ablation in patients before ablation. It also enables more precise management of patients after ablation, including the duration of antiarrhythmic drug therapy, monitoring for asymptomatic recurrence. Previous studies have shown that dilated LA increases the risk of AF recurrence after single ablation [31,32]. In our study, we also found the LAd ≥ 4.5 cm and AF duration ≥ 24.5 months were the recurrence predictors after PVI-plus linear ablation. 

## 5. Conclusions

In this study, the strategy of ablation (PVI plus linear ablation of mitral isthmus and posterior box isolation) was feasible and safe for persistent AF patients. Compared with the PVI-alone method, it improved outcomes in patients with persistent AF.

## 6. Limitations

Although this study reveals this strategy of ablation was feasible and safe, it has some limitations.

First, it was a retrospective design of study, and prospective randomized controlled studies and future studies with larger samples are needed to further validate it.

Although ADDs discontinuation after the blanking period was strongly recommended in our study, a beta-blocker was used in 18 (11.1%) and 20 (10.0%) patients combined with hypertension or coronary artery disease from the PVI and PVI-plus groups after the blanking period, which would have some impact on ablation recurrence rates.

## Figures and Tables

**Figure 1 jcm-12-00619-f001:**
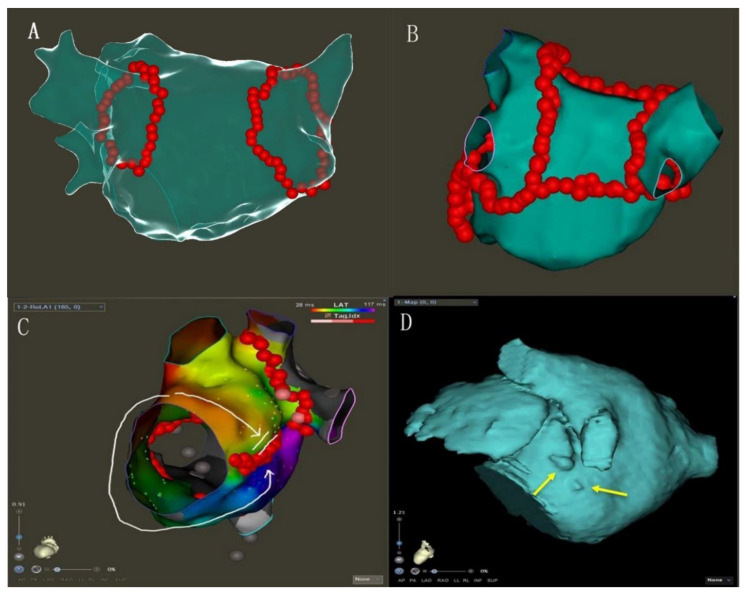
(**A**): Circumferential isolation of both ipsilateral pulmonary veins. (**B**): Typical recording of PVI-plus linear ablation of MI, LA roof and posterior inferior (posterior box isolation). (**C**): Conduction block of MI ablation line was examined by activation mapping: Mitral isthmus ablation line with bidirectional block (white arrow). (**D**): Typical MI pouches (indicated by yellow arrows) were observed in three-dimensional anatomical model.

**Figure 2 jcm-12-00619-f002:**
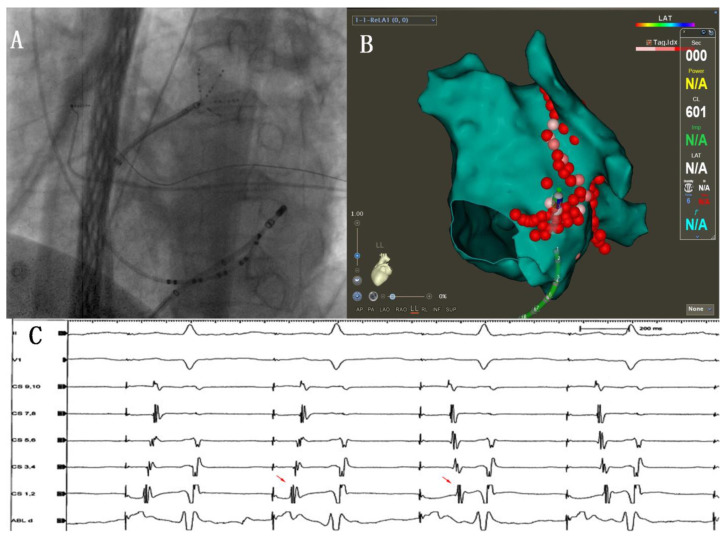
(**A**): Ablation was performed within the CS vein. (**B**): CS vein ablation: ablation catheter vector indicating endocardial plane. (**C**): Left atrial appendage pacing and the results in proximal-to-distal activation in the adjacent CS (as in sinus rhythm) during conduction block while being distal-to-proximal with persistent conduction (red arrows).

**Figure 3 jcm-12-00619-f003:**
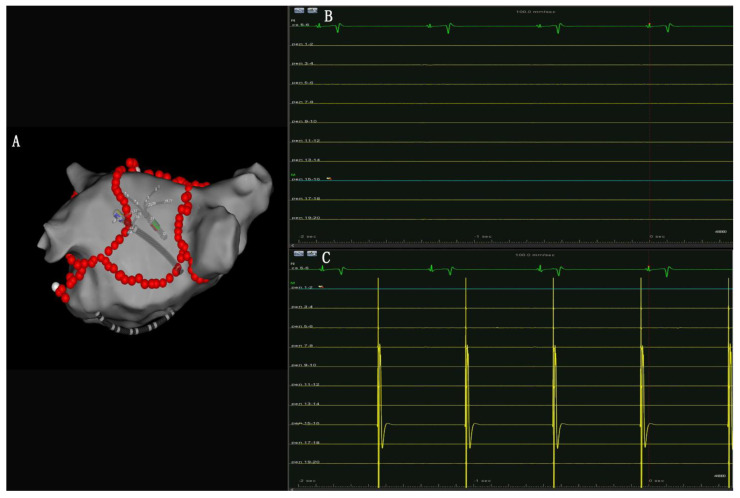
(**A**): Pentaray catheter was placed in LA posterior wall to assess PWI. (**B**): No atrial electrical signal was recorded by Pentaray catheter indicated a successful entry block of LA posterior wall. (**C**): LA posterior wall was paced using Pentaray catheter and failed 1:1 atrial capture was observed, indicating a successful enter block of LA posterior wall.

**Figure 4 jcm-12-00619-f004:**
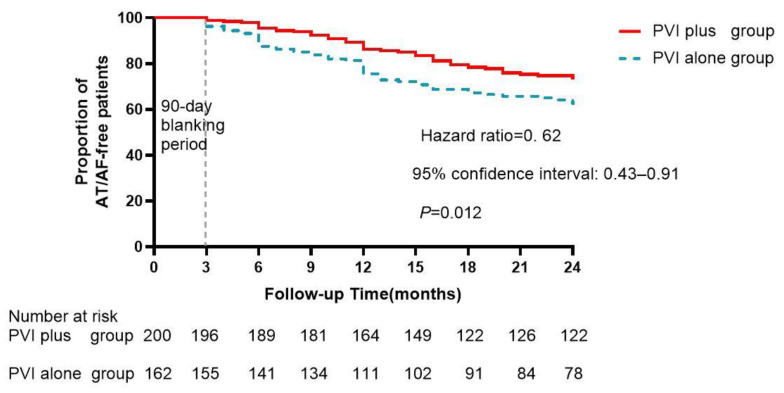
Atrial fibrillation (AF)/atrial tachycardia (AT)-free survival curve after index catheter ablation procedure.

**Figure 5 jcm-12-00619-f005:**
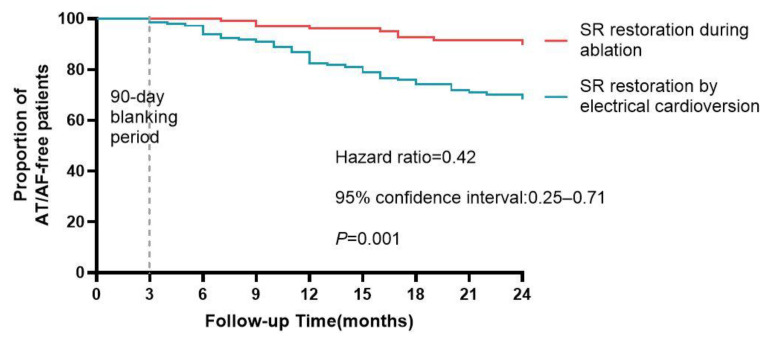
The AF/AT-free survival curves of persistent AF patients restored SR by catheter ablation or electrical cardioversion.

**Table 1 jcm-12-00619-t001:** The baseline characteristics of persistent AF patients.

Clinical Characteristics	PVI-Plus Group	PVI-Alone Group	*p* Value
Age (years)	58 ± 10	59 ± 8	0.222
Male	146 (73%)	110 (68%)	0.389
Duration of persistent AF (months)	28 ± 25	30 ± 24	0.556
Hypertension	92 (46%)	83 (51%)	0.323
Diabetes mellitus	21 (10.5%)	24 (15%)	0.162
Coronary artery disease	40 (20%)	39 (24%)	0.352
Prior stroke/TIA	69 (34.5%)	65 (40%)	0.359
CHA2DS2-VASc score	2.0 ± 1.7	2.2 ± 1.6	0.368
OAC	196 (98%)	160 (98%)	0.568
AADs therapy	136 (68%)	107 (66%)	0.437
Echocardiography parameters			
LVEF (%)	54 ± 6	52 ± 5	0.218
LAd (mm)	45 ± 3.8	45 ± 3.7	0.736

Values are presented as mean ± SD or as *n* (%). TIA, transitory ischemic attack. OAC, oral anticoagulation. AADs, antiarrhythmic drugs CHA2DS2-VASc, congestive heart failure, hypertension, age ≥ 75 (doubled), diabetes, stroke (doubled), vascular disease, age 65 to 74 and sex category (female). LAd, left atrial anteroposterior diameter.

**Table 2 jcm-12-00619-t002:** Procedural Data.

	PVI-Plus Group (*n* = 200)	PVI-Alone Group (*n* = 162)	*p* Value
Total procedure time, min	154 ± 21	83 ± 10	<0.001
Fluoroscopy time, min	19 ± 4	10 ± 2	<0.001
Successful PVI	200 (100%)	162 (100%)	-
MI ablation	200 (100%)	-	-
Successful MI block achieved	179 (89.5%)	-	-
CS vein ablation	152 (76%)	-	-
Posterior wall isolation	192 (96%)	-	-
Conversion to sinus rhythm	56 (28%)	13 (8%)	<0.001

PVI, pulmonary veins isolation. MI, mitral isthmus. CS, coronary sinus.

**Table 3 jcm-12-00619-t003:** Adverse Event.

Adverse Event	PVI-Plus Group (*n* = 200)	PVI-Only Group (*n* = 162)	*p* Value
Patients with adverse events *n* (%)	5 (2.5)	3 (1.9)	0.808
Death, *n* (%)	0	0	-
Symptomatic stroke, *n* (%)	1	0	0.648
Cardiac tamponade, *n* (%)	0	0	-
Phrenic nerve paralysis, *n* (%)	0	0	-
Femoral arteriovenous fistula	0	0	-
Femoral pseudoaneurysm	4	3	0.924
Atrioesophageal fistula, *n* (%)	0	0	-
Infection, *n* (%)	0	0	-
Heart failure, *n* (%)	0	0	-
Pulmonary vein stenosis	0	0	-

**Table 4 jcm-12-00619-t004:** Predictors of recurrence at follow-up in Cox regression analysis.

	Univariate	Multivariable
Variable	HR (95% CL) *p*	HR (95% CL) *p*
Age(years)	1.01 (0.98–1.04) 0.580	-
Female sex	0.91 (0.68–1.22) 0.518	-
AF duration	1.02 (1.01–1.02) 0.001	1.01 (1.00–1.02) 0.012
Hypertension	0.94 (0.54–1.64) 0.832	-
Diabetes mellitus	0.91 (0.36–2.29) 0.841	-
Coronary artery disease	0.87 (0.43–1.79) 0.711	-
Cardiac insufficiency	1.33 (0.42–4.28) 0.630	-
LA diameter	3.19 (1.53–6.67) 0.002	2.52(1.22–5.20) 0.012
CHA2DS2VASC score	1.04 (0.89–1.22) 0.637	-
No conversion to SR	2.61 (1.17–5.79) 0.019	2.45(1.10–5.44) 0.028

## Data Availability

The datasets used and/or analyzed during the current study available from the corresponding author on reasonable request.

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
