# Peer review of "Efficacy and Safety Ablation Index-Guided High-Energy Linear Ablation for Persistent Atrial Fibrillation: PVI Plus Linear Ablation of Mitral Isthmus and Posterior Box Isolation"

_jcm, 2023, doi:10.3390/jcm12020619_

Round 1
Reviewer 1 Report
Thank you for inviting me to review this manuscript. The Authors provided an interesting analysis on the efficacy of an extensive ablation procedure including PVI, PWI and MI in patients with persistent AF compared to a PVI only strategy. I have appreciated the provided figures that helps the reader to directly visualize treatment strategies and procedural aspects. On the contrary, the manuscript needs a careful check of language correctness.
In my opinion the topic of his review could be of interest for the readers of the Journal. However, there are some major revisions to be made prior to acceptance:
- Please, specify in the Method section that this is a retrospective analysis. In addition, in the limitation section specify each of the possible consequences of such study design and whether bias could have occurred in the selection of the ablation strategy.
- Could the Author specify the number of patients in each group that previously underwent other treatment strategies, such as rate control or rhythm control with AADs and electrical cardioversions? This could be useful to better understand the moment at which ablation procedure has been performed along AF history.
- "Discontinuation of antiarrhythmic drugs within the blanking period was strongly recommended". Could the Author report the number of patients that have been discharged with AADs after ablation procedure in each group and specify how long did the patients continued AAD therapy? Is the ablation performed during AAD assumptions?
- There are some concerns regarding statistical analysis. Continuous data should ne tested for normal distribution first, then should be presented as mean and standard deviation or as median and interquartile range, as appropriate. Finally, please specify how you found cut-off values for predictors of AF recurrence (i.e. Youden index).
- Could the Authors report other baseline characteristics that could potentially affect rhythm control efficacy (i.e. BMI, smoke, dyslipidemia, obstructive sleep apnea, physical inactivity)?
- How did you perform the brackthroughs EGM? What Rf parameters? How long? How patients?
- How did you prevent esophageal damage? Temperature probe? Intracardiac visualization?
8. The cutoff value of LAd predicting recurrence was 209 4.5 cm (AUC =0.61, P=0.023): what value of LAd? Before or after catheter ablation? What is the relationship between LAd and recurrence? Insert a comment and current literature position.
9. How did you perform a reinducition of AF? Burst? Isoproterenol infusion? Adenosine to test PV or atrial conduction breaktroughs?
I am looking forward to read you answers and revised manuscript.
Reviewer 2 Report
Xi Li MD et al. assessed the feasibility and clinical success rate of pulmonary veins isolation (PVI) combined with linear ablation of LA roof and posterior inferior (posterior wall isolation) and mitral isthmus compared with the PVI alone method in patients of persistent AF. They found that the ablation strategy of ablation (PVI plus linear ablation of mitral isthmus and posterior box isolation) was feasible and safe for persistent AF patients. Compared with the PVI alone method, it improved outcomes in patients with persistent AF.
Abstract: The authors write: "PVI combined 22 with linear ablation of left atrial posterior wall isolation(LAPWI)and MI were performed in PVI 23 plus group". This is not correct, as the authors did not aim for posterior wall isolation: For isolation you need ext block, not only delayed conduction.
--> The authors should write posterior box lesion insetad of isolation
Table 2: Again LAPWI is not correct. Please provide correct number for entrance and exit block in the box lesion.
Round 2
Reviewer 1 Report
Ok the comments and the answer were well Addressed.